# Integrating Application Methods and Concentrations of Salicylic Acid as an Avenue to Enhance Growth, Production, and Water Use Efficiency of Wheat under Full and Deficit Irrigation in Arid Countries

**DOI:** 10.3390/plants12051019

**Published:** 2023-02-23

**Authors:** Nabil Mohammed, Salah El-Hendawy, Bazel Alsamin, Muhammad Mubushar, Yaser Hassan Dewir

**Affiliations:** 1Department of Plant Production, College of Food and Agriculture Sciences, King Saud University, Riyadh 11451, Saudi Arabia; 2Department of Agricultural, Faculty of Agriculture and Veterinary Medicine, Thamar University, Thamar 87246, Yemen; 3Department of Agronomy, Faculty of Agriculture, Suez Canal University, Ismailia 41522, Egypt; 4Horticulture Department, Faculty of Agriculture, Kafrelsheikh University, Kafr El Sheikh 33516, Egypt

**Keywords:** chlorophyll content, foliar spray, grain yield, multivariate analysis, relative water content, seed soaking

## Abstract

As water deficit in arid countries has already become the norm rather than the exception, water conservation in crop production processes has become very critical. Therefore, it is urgent to develop feasible strategies to achieve this goal. Exogenous application of salicylic acid (SA) has been proposed as one of the effective and economical strategies for mitigating water deficit in plants. However, the recommendations concerning the proper application methods (AMs) and the optimal concentrations (Cons) of SA under field conditions seem contradictory. Here, a two-year field study was conducted to compare the effects of twelve combinations of AMs and Cons on the vegetative growth, physiological parameters, yield, and irrigation water use efficiency (IWUE) of wheat under full (FL) and limited (LM) irrigation regimes. These combinations included seed soaking in purified water (S_0_), 0.5 mM SA (S_1_), and 1.0 mM SA (S_2_); foliar spray of SA at concentrations of 1.0 mM (F_1_), 2.0 mM (F_2_), and 3.0 mM (F_3_); and combinations of S_1_ and S_2_ with F_1_ (S_1_F_1_ and S_2_F_1_), F_2_ (S_1_F_2_ and S_2_F_2_), and F_3_ (S_1_F_3_ and S_2_F_3_). The results showed that the LM regime caused a significant reduction in all vegetative growth, physiological, and yield parameters, while it led to an increase in IWUE. The application of SA through seed soaking, foliar application, and a combination of both methods increased all of the studied parameters in all the evaluated times, resulting in higher values for all parameters than the treatment without SA (S_0_). The multivariate analyses, including principal component analysis and heatmapping, identified the foliar application method with 1–3 mM SA alone or in combination with seed soaking with 0.5 mM SA as the best treatments for the optimal performance of wheat under both irrigation regimes. Overall, our results indicated that exogenous application of SA has the potential to greatly improve growth, yield, and IWUE under limited water application, while optimal coupling combinations of AMs and Cons were required for positive effects in field conditions.

## 1. Introduction

One of the greatest challenges to the sustainability of irrigated agriculture and food production in arid and semiarid regions of the world is the scarcity of freshwater. This challenge will become more pressing in these regions as the population continues to increase and the impact of climate change intensifies; in particular, current climate changes in these regions are associated with increments in mean and utmost temperatures and low precipitation simultaneously. Presently, global croplands consume 8266 km^3^ year^−1^ of water: 5406 km^3^ from green water and 2860 km^3^ from blue water [1]. The global irrigated croplands cover 23% of the global cropland areas, consume 1083 km^3^ year^−1^ of blue water resources, and provide approximately 40% of global food production [1,2]. Rosa et al. [1] reported that, with 20% to 50% irrigation deficit scenarios in currently irrigated land, it is possible to irrigate an additional 10% of global croplands and increase food production to feed an additional 800 million people. Therefore, applying deficit irrigation practices instead of full irrigation could be one reasonable solution for handling food crises caused by the freshwater shortage [3,4]. However, the exposure of plants to water-deficit stress, especially during critical growth stages, can result in reductions in dry matter production and final grain yields of more than 50% for the majority of cultivated crops [5,6,7,8]. This is because water-deficit stress causes a broad spectrum of adverse impacts on several physiological, morphological, and biochemical attributes that eventually impair the growth and development of plants, with significant decreases in their production. Exposure of plants to water-deficit stress quite often leads to substantial reductions in cell enlargement rate, biomass accumulation, leaf area, different yield components, photosynthesis rate, stomatal conductance, chlorophyll content (Chl), and relative water content (RWC) [8,9,10,11,12]. It also causes an increase in leaf temperature and the generation of reactive oxygen species (ROS) [13] and leads to the senescence of leaves due to the breakdown of chlorophyll [13,14]. In addition, it leads to an imbalance in several plant hormones and growth regulators [15,16]. Therefore, applying irrigation water below the full crop water requirements necessitates further complementary approaches to mitigate the harmful outcomes of water deficit on plant growth and production.

Recently, a number of different agronomic and physiological practices have been applied as complementary approaches in order to reduce water-deficit-induced crop losses. Fortunately, plants incubate several complex and well-organized mechanisms to mitigate the deleterious effects of different abiotic stressors. The ability of plants to biosynthesize and accumulate various compatible osmolytes is considered one of the most common of these mechanisms. Plants usually accumulate different compatible osmolytes under stressful conditions in order to maintain cell turgor, maintain continuous water uptake at low soil water potential, protect cellular machinery from stresses, remove excess levels of ROS, enhance the activities of antioxidant enzymes, and protect proteins and biological membranes [17,18].

Salicylic acid (SA), as an example of these osmolytes, is present in most plants; however, concentrations differ significantly between species [19]. For example, although the concentration of SA in tobacco leaves is less than 100 ng g^−1^ fresh weight, this concentration can reach 10 mg g^−1^ fresh weight in potato leaves [19]. Additionally, not all plants have the capacity to accumulate SA at a concentration level sufficient to contribute significantly to protecting themselves from the deleterious effects of water-deficit stress. Therefore, previous studies have suggested that external application of SA, either through seed and/or foliar treatments, is one of the most widely applied methods for elevating SA concentration to a sufficient level in plants that are unable to synthesize it under water-deficit stress [17,18,19]. Thus, the exogenous application of SA could be considered an easy and cost-effective approach for alleviating the harmful effects of water-deficit stress on the growth and production of plants.

After this initial characterization, several reports have elucidated the role of SA as a phytohormone and its vital contribution and multifaceted role in plants in enhancing their performance under abiotic stress conditions. There is much evidence that the exogenous application of SA at an appropriate concentration affects multiple aspects related to plant growth, development, and production under stress as well as normal conditions. Exogenous application of SA has also been found to enhance different physiological processes, such as photosynthetic activity, stomatal regulation, nutrient uptake and transport, Chl and protein synthesis, RWC, leaf water potential, and antioxidant capacity [7,18,20,21,22,23]. Plants treated with SA also showed a decrease in leaf senescence [13,14,24]. For example, Ilyas et al. [12] reported that seeds treated with 10 mM SA under water-deficit stress improved germination percentage by 21% and increased shoot length and leaf water potential by 20% and 47%, respectively, as compared with non-treated seeds. Kareem et al. [25] investigated the effect of spraying 1.44 and 2.88 mM of SA on wheat under water-deficit stress. They reported that SA enhanced growth, yield, and physiological traits, such as plant height, spike length, number of grains per spike, thousand-grain weight (TGW), Chl content, and RWC, with a concentration of 1.44 mM being shown to have the most positive impact compared with 2.88 mM. Azmat et al. [7] reported that wheat plants treated with 1.0 mM SA as a foliar application significantly increased Chl a, Chl b, and RWC by 125%, 167%, and 238% under drought stress, respectively, when compared with untreated plants. Hafez et al. [23] found that synergistic use of biochar and SA significantly improved several physico-agronomic traits, such as Chl content, RWC, photosynthetic rate, stomatal conductance, nutrient uptake (N, P, and K), number of grains spike^−1^, TGW, and GY, when compared with control treatments under water-deficit conditions.

Although the beneficial effects of SA on stress tolerance are relatively well known, doubts remain about the application method (foliar application or seed soaking) and the appropriate concentration of SA that would provide the best results against water-deficit stress. For instance, Korkmaz et al. [26] found that both seed soaking and foliar application of SA within the range of 0.1–1 mM provided similar means of protection for muskmelon plants against drought stress. However, seed soaking with up to 0.5 mM SA had a positive effect on all measured traits compared with foliar application with different concentrations or seed soaking with concentrations below 0.5 mM SA. Seed soaking with 100 ppm resulted in significantly higher wheat growth, yield components, and GY in wheat under conserved soil moisture conditions than a foliar application with the same concentration in the study of Mevada et al. [27]. Otherwise, Farooq et al. [28] demonstrated that foliar application of SA was more efficient than seed soaking at the same concentrations for enhancing photosynthesis and plant growth in rice against water deficiency. Additionally, they also found that foliar application with 100 mg L^−1^ SA was the best treatment to improve the performance of the rice plants under normal and stress conditions compared with 50 and 150 mg L^−1^ or the control treatment. Besides the application methods of SA, the concentration can also influence the ability of SA to mitigate the negative impacts of abiotic stresses on the growth and production of plants, as several plant functions can be inhibited or induced with high and low SA concentrations, respectively. Several studies reported that the application of SA at a relatively low concentration was more effective than a higher concentration in enhancing the growth, physiological, and productivity parameters of different field crops under various abiotic stresses. For example, Kang et al. [29] found that pretreatment seeds of wheat with 0.5 mM SA for 3 days significantly increased the height, fresh weight, and dry wheat of seedlings under drought stress by 10.7%, 15.4%, and 10.4%, respectively, as compared with the control treatment. However, these three traits in the pretreatment seeds with 3.0 mM SA were significantly lower than in the control (by 38.9%, 51.4%, and 36.2%, respectively). In another instance, 1.0 mM SA inhibited the growth of mung bean (*Vigna radiata* L.) under salinity stress; however, promoted photosynthesis and growth was evidenced with 0.1 and 0.5 mM SA [30]. In contrast, Sohag et al. [24] found that a concentration of 1.0 mmol L^−1^ SA equally improved the water-stress tolerance of rice seedlings, as did 0.5 mmol L^−1^. Soaking seeds of wheat with 10 mM SA has the potential to enhance the growth of plants under drought stress [12]. However, in the study of Parveen et al. [31], it was reported that the resistance of wheat plants to water-deficit stress improved when the plants were sprayed with relatively high concentrations of SA (3 and 6 mM SA). Moreover, 0.6 mM, 0.01–0.05 mM, and 0.1–0.5 mM were found to be appropriate concentrations for enhancing the growth, production, and antioxidant defense mechanisms of soybean, wheat, and bean crops, respectively, under deficit irrigation when their seeds were treated with these concentrations [32,33,34]. 

Therefore, we hypothesized that the method of application, concentration, level of stress, and crop type are the main factors that determine the role of SA in enhancing the growth and yield characteristics of wheat, particularly under field conditions. Therefore, the principal objective of this study was to identify the proper application methods for SA along with an effective concentration to enhance the vegetative growth and yield characteristics and water use efficiency of wheat under FL and LM irrigation regimes in an arid agro-ecosystem.

## 2. Results

### 2.1. Vegetative Growth Parameters

Based on ANOVA analysis, irrigation regimes (IRs) and SA treatments had significant impacts on all vegetative growth parameters at both sampling dates (80 and 100 DAS) and in the two growing seasons, except for PH and TN, which showed non-significant variation between the SA treatments at the first sampling in the first season and at the second sampling in both seasons (Table 1). The IR-by-SA interaction had a significant effect on all vegetative growth parameters, except PH and TN (Table 1).

The LM regime resulted in significant reductions in all vegetative growth parameters in both samples and growing seasons when compared with the FL regime. Averaged over the two seasons, the LM regime decreased PH, TN, GLN, GLA, SFW, and SDW by 15.3%, 12.6%, 25.9%, 34.2%, 23.8%, and 18.8% at 80 DAS and by 15.8%, 17.0%, 31.4%, 39.3%, 30.0%, and 33.0% at 100 DAS, respectively, when compared with the FL regime (Table 2). 

Regardless of the irrigation regimes, the application of SA considerably improved all vegetative growth parameters in comparison with non-treated plants (S_0_), except PH and TN, as indicated in Table 3. However, foliar application of SA with different concentrations alone (F_1_, F_2_, and F_3_) or in combination with seed soaking with 0.5 mM (S_1_F_1_, S_1_F_2_, and S_1_F_3_) proved more effective in enhancing the different growth parameters than seed soaking alone (S_1_ and S_2_) or a combination of foliar application with seed soaking with 1.0 mM (S_2_F_1_, S_2_F_2_, and S_2_F_3_; Table 3). Averaged over the two seasons, the former six treatments increased the different growth parameters by 1.6–16.8% at 80 DAS and by 0.7–24.1% at 100 DAS, when compared with the latter five treatments. Additionally, the former six and latter five treatments increased the different growth parameters by 4.4–22.7% and 2.9–9.9% at 80 DAS and by 4.7–36.9% and 4.0–16.8% at 100 DAS, respectively, when compared with the S_0_ treatments (Table 3).

Figure 1 shows the responses of vegetative growth parameters to different SA treatments under each irrigation regime. Generally, in both irrigation regimes, the values of GLN, GLA, SFW, and SDW were significantly higher for the three foliar treatments (F_1_, F_2_, and F_3_) as well as the combination of these three treatments with S_1_ (S_1_F_1_, S_1_F_2_, and S_1_F_3_) than for the other SA treatments and the S_0_ treatment. Averaged over the two seasons, the three foliar treatments resulted in increases in the four aforementioned vegetative growth parameters of 11.3–18.5% and 10.3–27.8% under the FL regime and of 21.9–29.4% and 15.0–50.2% under the LM regime at 80 and 100 DAS, respectively, compared with the S_0_ treatment.

The three combination treatments (S_1_F_1_, S_1_F_2_, and S_1_F_3_) resulted in increases in the four aforementioned vegetative growth parameters of 14.4–20.5% and 9.1–28.1% under the FL regime and of 18.2–30.5% and 18.8–52.8% under the LM regime at 80 and 100 DAS, respectively, compared with the S_0_ treatment (Figure 1). Under the FL regime, some vegetative growth parameters tended to decrease in the seed soaking treatment with 1.0 mM SA (S_2_), while this treatment increased these vegetative growth parameters under the LM regime by 4.4–18.3% and 5.5–36.4% at 80 and 100 DAS, respectively, compared with the S_0_ treatment (Figure 1). 

### 2.2. Physiological Parameters

The two physiological parameters (RWC and Chlt) measured at 100 DAS were significantly affected by treatments of IR and SA as well as their interaction (Table 1). Generally, both parameters were superior in the FL regime compared to the LM regime. Over the two seasons, the values of RWC and Chlt were lower (by 12.8% and 29.0%, respectively) under the LM regime compared to the FL regime (Table 2). Both parameters were also significantly affected by the SA treatments regardless of the IR treatments. The highest values for RWC were observed when the plants were treated with the three foliar treatments alone as well as the combination of the foliar treatments with S_1_. The highest values of Chlt were observed when the plants were treated with F_2_ and S_1_F_2_, whereas the values of both parameters for seed soaking (S_1_ and S_2_) and the combination of foliar treatments and S_2_ (S_2_F_1_, S_2_F_2_, and S_2_F_3_) were statistically at par with the S_0_ treatment (Table 3).

The response of both parameters to the different SA treatments was also dependent on the irrigation regime (Figure 2). Under the FL regime, the values of RWC for all SA treatments were statistically at par with the S_0_ treatment, with the exception of S_2_F_1_, S_2_F_2_, and S_2_F_3_, the values of RWC for these treatments being 2.1% and 4.8% lower than their corresponding values for the S_0_ treatment in the first and second seasons, respectively. The highest values of Chlt under the FL regime were obtained with F_2_ and S_1_F_2_, followed by F_1_ and S_1_F_1_, while the other SA treatments were statistically at par with the S_0_ treatment (Figure 2). Under the LM regime, F_2_, F_3_, and S_1_F_2_ showed maximum increases in RWC of about 9.5%, and F_2_, S_1_F_2_, S_2_F_1_, and S_2_F_2_ presented the highest Chlt values (by 17.8–22.5%), as compared with S_0_ treatment. Additionally, the values for Chlt for S_1_, S_2_, and S_2_F_3_ were statistically at par with the S_0_ treatment, while these treatments showed higher RWC values compared with the S_0_ treatment (by 2.8–8.4%) (Figure 2).

### 2.3. Yield Parameters and Irrigation Water Use Efficiency

The different yield parameters and IWUE were significantly affected by the IR and SA treatments, except SpNS, which showed non-significant variation among the SA treatments in the first season (Table 4). The IR-by-SA interaction also had significant effects on GWS, GY, HI, and IWUE in both growing seasons (Table 4). 

Averaged over the two seasons, the LM regime decreased SL, SpNS, GNS, GWS, TGW, GY, BY, and HI by 6.7%, 3.9%, 9.8%, 19.0%, 10.2%, 37.8%, 29.4%, and 12.1%, respectively, while it increased IWUE by 19.7%, when compared with the FL regime (Table 5). 

Regardless of the irrigation regimes, the different SA treatments considerably improved all yield parameters and IWUE in comparison with the S_0_ treatment, except SpNS in the first season, as presented in Table 5. The highest values for all yield parameters and IWUE were noted for the three foliar treatments alone as well as the combination of foliar treatments with S_1_. Averaged over two seasons, these treatments increased SL, SpNS, GNS, GWS, TGW, GY, BY, HI, and IWUE by 5.0%, 2.1%, 6.3%, 10.4%, 6.1%, 23.9%, 19.5%, 7.6%, and 28.6%, respectively, when compared with the S_0_ treatment. The other SA treatments, especially the seed soaking treatments (S_1_ and S_2_) as well as the combination of foliar treatments with S_2_, also increased the most yield parameters and IWUE, as compared with the S_0_ treatment, but at lower rates than the previous treatments. Averaged over two seasons, these treatments increased SL, SpNS, GNS, GWS, TGW, GY, BY, HI, and IWUE by 3.3%, 0.7%, 2.7%, 3.6%, 2.9%, 11.2%, 8.5%, 4.8%, and 15.7%, respectively, when compared with the S_0_ treatment (Table 5).

Figure 3 shows the responses of IWUE and yield parameters to different SA treatments under each irrigation regime. Generally, in both irrigation regimes, IWUE and yield parameters (GWS, GY, and HI) were highest with a foliar application of SA for both growing seasons, followed by combinations of foliar treatments and seed soaking with 0.5 mM. Additionally, the values of GWS, GY, HI, and IWUE for both treatments of seed soaking (S_1_ and S_2_) were statistically at par with the S_0_ treatment under the FL regime, while both treatments showed higher values for these parameters compared with the S_0_ treatment (by 7.1–28.0%) under the LM regime. Similarly, the values of GWS, GY, HI, and IWUE for the S_2_F_1_, S_2_F_2_, and S_2_F_3_ treatments were statistically at par with the S_0_ treatment under the FL regime, while they showed higher values for these parameters (by 8.0–26.7%) compared with the S_0_ treatment under the LM regime (Figure 3).

### 2.4. Relationship between Different Parameters under the Full and Limited Irrigation Regimes

The Pearson’s correlation analysis between and among the various vegetative growth, physiological, yield, and IWUE parameters under the FL and LM regimes over two seasons is presented in Table 6. All vegetative growth parameters measured at 80 and 100 DAS showed strong and positive correlations with each other under both irrigation regimes (r = 0.53–0.93), with the exception of PH at 100 DAS, which had non-significant correlations with the other vegetative growth parameters under the FL regime. Both physiological parameters (RWC and Chlt) had non-significant correlations with all vegetative growth parameters under the FL regime, while they showed strong (r = 0.64–0.93) and moderate (r = 0.61–0.74) correlations, respectively, with vegetative growth parameters under the LM regime (Table 6). Nearly all yield parameters and IWUE showed positive and strong correlations with each other and with vegetative growth parameters under both irrigation regimes, with a few exceptions under the FL regime. Under the FL regime, RWC showed a non-significant correlation while Chlt showed a moderate correlation (r = 0.60–0.67) with yield parameters and IWUE. However, under the LM regime, RWC and Chlt showed strong (r = 0.68–0.95) and moderate (r = 0.58–0.75) correlations, respectively, with yield parameters and IWUE (Table 6).

### 2.5. Multivariate Approach: An Overview of the Responses of Plant Parameters to Different SA Treatments

The responses of different parameters of wheat to the SA treatments under the FL and LM conditions were represented by a biplot of a principal component analysis (PC) (Figure 4) and heatmap clustering (Figure 5). The PCA biplot shows that the first two components accounted for 76.67% and 81.99% of the total variability between all parameters under the FL and LM conditions, respectively (Figure 4). The first component (PC1) accounted for 70.67% and 75.26%, and the second component (PC2) accounted for 6.00% and 6.72% of the total variability between all parameters under the FL and LM conditions, respectively. Additionally, the biplot of the PCA shows that the different SA treatments were separated into four groups and that the responses of various parameters to these groups were almost similar under both irrigation regimes. The first and second groups included the three foliar treatments and the combination of the three foliar treatments with S_1_. However, the third and fourth groups included the control treatment (S_0_), both soaking treatments (S_1_ and S_2_), and the combination of S_2_ with the three foliar treatments. The first and second groups were situated along the quarters with the highest PC1 and lowest or highest PC2, and they were closely correlated with all parameters. However, the third and fourth groups were situated along the quarters with the lowest PC1 and lowest or highest PC2, and they did not show any relationship with different parameters (Figure 4).

Similarly, the heatmap cluster analysis based on all the parameters divided the SA treatments into two and three main groups under the FL and LM regimes, respectively (Figure 5). The three foliar treatments alone or in combination with S_1_ were clustered into one group and displayed the highest values for most parameters under both irrigation regimes. However, the other SA treatments were clustered with the S_0_ treatment into one group under the FL regime and into two groups under the LM regime, while the S_0_ treatment was clustered alone into one group (Figure 5). 

**Table 6 plants-12-01019-t006:** Pearson’s correlation matrix for different parameters (Par.) of vegetative growth at 80 and 100 days from sowing, physiological, yield, and IWUE of wheat across two seasons under full irrigation (lower left) and limited irrigation (upper right) regimes.

Par.	1	2	3	4	5	6	7	8	9	10	11	12	13	14	15	16	17	18	19	20	21	22	23
PH-1 (1)		**0.68**	**0.81**	**0.76**	**0.71**	**0.66**	**0.86**	**0.78**	0.51	**0.79**	0.45	**0.63**	**0.82**	0.36	**0.60**	**0.68**	**0.69**	**0.64**	**0.66**	**0.75**	**0.76**	0.58	**0.75**
TN-1 (2)	**0.73**		0.53	**0.87**	**0.71**	**0.71**	0.54	**0.81**	**0.87**	**0.93**	**0.59**	**0.80**	**0.83**	**0.61**	0.52	**0.74**	**0.91**	**0.83**	**0.66**	**0.84**	**0.82**	**0.71**	**0.84**
GLN-1 (3)	**0.65**	**0.61**		**0.80**	**0.85**	**0.80**	**0.63**	**0.61**	**0.66**	**0.70**	**0.68**	**0.67**	**0.83**	**0.65**	**0.78**	**0.74**	**0.69**	**0.71**	**0.73**	**0.79**	**0.85**	0.45	**0.79**
GLA-1 (4)	**0.64**	**0.79**	**0.72**		**0.90**	**0.84**	**0.60**	**0.74**	**0.90**	**0.90**	**0.66**	**0.83**	**0.93**	**0.65**	**0.70**	**0.79**	**0.94**	**0.96**	**0.86**	**0.96**	**0.97**	**0.68**	**0.96**
SFW-1 (5)	**0.64**	**0.86**	**0.71**	**0.80**		**0.94**	0.56	0.58	**0.73**	**0.83**	**0.62**	**0.84**	**0.90**	**0.71**	**0.88**	**0.88**	**0.84**	**0.86**	**0.85**	**0.88**	**0.89**	**0.63**	**0.88**
SDW-1 (6)	**0.64**	**0.93**	**0.67**	**0.72**	**0.94**		0.36	0.48	**0.69**	**0.80**	**0.59**	**0.79**	**0.86**	**0.74**	**0.85**	**0.93**	**0.75**	**0.78**	**0.72**	**0.80**	**0.82**	0.53	**0.80**
PH-2 (7)	**0.64**	0.42	0.17	0.21	0.45	0.54		**0.73**	**0.60**	**0.65**	**0.58**	**0.58**	**0.64**	0.18	0.48	0.41	**0.63**	0.56	**0.65**	**0.67**	**0.63**	**0.68**	**0.67**
TN-2 (8)	**0.58**	**0.77**	**0.58**	**0.58**	**0.75**	**0.80**	0.36		**0.68**	**0.79**	0.35	**0.59**	**0.69**	0.13	0.45	**0.60**	**0.72**	**0.60**	0.49	**0.68**	**0.70**	0.49	**0.68**
GLN-2 (9)	**0.60**	**0.76**	**0.70**	**0.83**	**0.74**	**0.70**	0.01	**0.59**		**0.84**	**0.59**	**0.71**	**0.78**	**0.66**	0.46	**0.61**	**0.89**	**0.91**	**0.73**	**0.87**	**0.88**	**0.63**	**0.87**
GLA-2 (10)	**0.74**	**0.87**	**0.60**	**0.82**	**0.88**	**0.92**	0.54	**0.74**	**0.78**		**0.71**	**0.88**	**0.93**	**0.65**	**0.67**	**0.80**	**0.90**	**0.87**	**0.79**	**0.92**	**0.90**	**0.77**	**0.92**
SFW-2 (11)	**0.71**	**0.74**	**0.62**	**0.76**	**0.73**	**0.76**	0.43	**0.77**	**0.72**	**0.91**		**0.83**	**0.71**	**0.65**	0.44	0.45	**0.60**	**0.65**	**0.68**	**0.68**	**0.66**	**0.62**	**0.68**
SDW-2 (12)	**0.78**	**0.84**	**0.67**	**0.84**	**0.83**	**0.80**	0.39	**0.80**	**0.85**	**0.90**	**0.91**		**0.91**	**0.66**	**0.73**	**0.80**	**0.81**	**0.84**	**0.84**	**0.84**	**0.84**	**0.65**	**0.84**
RWC-2 (13)	0.47	0.48	0.17	0.56	0.42	0.54	0.12	0.40	0.50	0.53	0.56	0.52		**0.68**	**0.74**	**0.85**	**0.88**	**0.90**	**0.90**	**0.94**	**0.95**	**0.68**	**0.94**
Chlt-2 (14)	0.39	0.54	0.23	0.50	0.55	0.57	0.06	0.53	0.45	0.55	0.54	0.53	0.42		**0.58**	**0.58**	**0.60**	**0.75**	**0.71**	**0.71**	**0.71**	**0.63**	**0.71**
SL (15)	0.56	0.51	0.35	0.56	0.36	0.44	0.18	0.47	0.48	**0.64**	**0.74**	0.54	0.38	**0.61**		**0.87**	**0.67**	**0.63**	**0.69**	**0.71**	**0.69**	**0.60**	**0.71**
SpNS (16)	0.42	**0.62**	0.42	**0.73**	0.53	0.56	0.27	0.56	0.49	**0.69**	**0.70**	**0.63**	0.55	**0.60**	**0.67**		**0.75**	**0.69**	**0.64**	**0.74**	**0.76**	0.51	**0.74**
GNS (17)	**0.78**	**0.83**	0.47	**0.78**	**0.81**	**0.77**	0.26	**0.71**	**0.79**	**0.83**	**0.86**	**0.92**	0.53	**0.63**	**0.58**	0.51		**0.95**	**0.83**	**0.96**	**0.93**	**0.83**	**0.96**
GWS (18)	**0.79**	**0.93**	0.47	**0.77**	**0.83**	**0.86**	0.41	**0.73**	**0.71**	**0.86**	**0.83**	**0.88**	0.50	**0.60**	0.58	**0.63**	**0.94**		**0.93**	**0.98**	**0.97**	**0.77**	**0.98**
TGW (19)	**0.66**	**0.91**	0.45	**0.74**	**0.83**	**0.88**	0.46	**0.65**	0.54	**0.82**	**0.62**	**0.69**	0.56	**0.67**	0.47	**0.67**	**0.71**	**0.86**		**0.92**	**0.90**	**0.76**	**0.92**
GY (20)	**0.70**	**0.85**	**0.60**	**0.86**	**0.87**	**0.88**	0.46	**0.72**	**0.75**	**0.95**	**0.93**	**0.90**	0.52	**0.64**	**0.64**	**0.76**	**0.87**	**0.91**	**0.80**		**0.98**	**0.82**	**1.00**
BY (21)	**0.71**	**0.83**	**0.62**	**0.89**	**0.87**	**0.84**	0.44	**0.65**	**0.74**	**0.92**	**0.89**	**0.87**	0.49	**0.60**	**0.63**	**0.78**	**0.86**	**0.90**	**0.80**	**0.99**		**0.71**	**0.98**
HI (22)	**0.62**	**0.81**	0.50	**0.71**	**0.76**	**0.85**	0.44	**0.80**	**0.74**	**0.92**	**0.94**	**0.88**	0.50	**0.65**	**0.60**	**0.60**	**0.82**	**0.83**	**0.69**	**0.91**	**0.82**		**0.82**
IWUE (23)	**0.70**	**0.85**	**0.60**	**0.86**	**0.87**	**0.88**	0.46	**0.72**	**0.75**	**0.95**	**0.93**	**0.90**	0.52	**0.64**	**0.64**	**0.76**	**0.87**	**0.91**	**0.80**	**1.00**	**0.99**	**0.91**	

PH, plant height (cm plant^−1^); TN, tiller number per plant; GLN, green leaf number per plant; GLA, green leaf area (cm^2^ plant^−1^); SFW, shoot fresh weight (g plant^−1^); SDW, shoot dry weight (g plant^−1^); RWC, relative water content (%); Chlt, total chlorophyll content (mg g^−1^ fresh weight), SL, spike length (cm); SpNS, spikelet number per spike; GNS, grain number per spike; GWS, grain weight per spike (g); TGW, thousand-grain weight (g); GY, grain yield (ton ha^−1^); BY, biological yield (ton ha^−1^); HI, harvest index (%); IWUE, irrigation water use efficiency (kg mm^−1^ ha^−1^). The numbers 1 and 2 indicate traits measured at 80 and 100 days after sowing, respectively. Values in bold indicate a significance level alpha = 0.05.

## 3. Discussion

Generally, exposure of wheat plants to water-deficit stress causes drastic reductions in their growth attributes and productivity, which may exceed 50–70%, especially if this water deficit coincides with critical growth stages [4,35,36]. In this study, which was conducted under arid conditions, the LM regime consistently resulted in lower growth and production of wheat compared to the FL regime, while the opposite was true for IWUE (Table 2 and Table 5).

These significant reductions in different parameters under the LM regime may be due to the fact that water-deficit stress triggers a broad spectrum of adverse impacts on several physiological, morphological, and biochemical attributes that eventually impair the growth and development of plants, with significant decreases in their production. Water-deficit stress quite often leads to substantial inhibition of several morpho-physiological attributes of plants. It leads to the inhibition of cell division, cell expansion, gas-exchange rates, stomatal conductance, biomass accumulation, and leaf area. It also causes an imbalance in several plant hormones and growth regulators; increases leaf temperature, which reduces RWC and increases transpiration rates; reduces the uptake and translocation of macronutrients; and induces oxidative damage, which may affect leaf water status and leaf pigments [8,9,10,11,12,15,16,35,37,38]. Consequently, decrease in wheat growth, yield, and yield components is the LM regime’s expected outcome. Therefore, it was difficult to apply the LM regime to wheat without an accompanying reduction in growth and production. Thus, wheat production under the LM regime requires further strategies to decrease the deleterious effects of the shortage of water on wheat growth and production.

Currently, exogenous application of phytohormones such as SA through seed soaking and/or foliar application has been considered an efficient, easy, and economic strategy for ameliorating the harmful effects of water-deficit stress on the growth and production of field crops. This may be because the exogenous application of SA plays an important role in modulating and inducing several biochemical and physiological mechanisms to control different plant responses under stress as well as normal conditions [19,31,39,40,41,42]. In this study, treatments with SA through seed soaking (S_1_ and S_2_), foliar application (F_1_, F_2_, and F_3_), and a combination of both methods (S_1_F_1_–S_2_F_3_) exhibited significant increases in all studied parameters in comparison with the untreated condition (S_0_) (Table 3 and Table 5). These results indicate that the exogenous application of SA could represent an alternative and technically simple eco-friendly approach to stabilize the growth and production of plants under either stress or normal conditions. These results are in agreement with previous studies showing that the exogenous application of SA plays an important role in enhancing the growth, chlorophyll content, RWC, GY, yield components, and water use efficiency of different field crops under normal and stress conditions [16,31,34,43,44,45]. These findings may be attributed to SA’s being a phenol-based phytohormone that is involved in a wide range of developmental, biochemical, and physiological processes under different growth conditions. It plays a vital role in enhancing cell division, cell elongation, photosynthetic activity, stomatal opening, chlorophyll pigment contents, nutrient uptake, biomass accumulation, and RWC. It also plays an integrating role in delaying the senescence of plant organs and regulating the source-to-sink relationship, as well as in plant growth and development pathways and some physiological responses related to carbon uptake and/or fixation in the chloroplasts and Rubisco concentration and activity [20,39,41,42,45]. These abovementioned advantages of SA might explain why the plants treated with SA, in general, exhibited greater enhancements in growth, physiological attributes, yield components, yield, and IWUE. The ability of SA to delay the senescence of plant organs and avoid the premature loss of anthesis and grains leads to increments in yield-related traits in wheat [31]. The ability of SA to increase the availability of CO_2_ for photosynthesis, as well as increase leaf diffusive resistance and decrease transpiration rates by regulating stomatal opening and closing, leads to improved IWUE [46,47]. The ability of SA to enhance the flow of metabolites to developing grains leads to improved TGW and thus improved GY [8]. Additionally, because SA is involved in the formation of antioxidants, stimulating osmotic adjustment, and scavenging ROS, these mechanisms lead to the protection of membrane integrity and photosynthetic pigments, which ultimately lead to enhanced growth characteristics of plants [20,48,49]. These positive effects of SA may provide a rationale for the importance of exogenous application of SA in improving the growth-, physiological-, and yield-related parameters considered in this study, given the positive effects of the SA treatments observed in comparison with the S_0_ treatment. Additionally, the enhancement of growth and physiological attributes by the application of SA was always linked to the high production and IWUE of the crop, which was confirmed in this study through significant and positive correlations of growth and physiological parameters with yield parameters and IWUE under the LM regime and largely under the FL regime (Table 6). Additionally, the strong correlations of RWC and Chlt with the different parameters of growth and yield as well as IWUE under the LM regime (Table 6) indicated the involvement of SA in the protection of photosynthetic pigments by reducing oxidative stress and improved leaf RWC via the lowering of leaf water potential and the accumulation of compatible solutes, which allows additional water to be taken up from the soil, especially under water-deficit conditions.

It appears from previous studies that the efficiency of the exogenous application of SA for enhancement of the growth and production of crops and the alleviation of the negative effects of abiotic stresses depended on several aspects, including plant species, concentrations, application methods, application time, stress duration, and also the physiological state of the plant. Among these aspects, the identification of the proper concentrations and the appropriate application methods for SA are the two factors that should be investigated to avoid unanticipated results from the exogenous application of SA. In this study, in general, foliar application of SA was more effective than seed soaking for improving growth-, physiological-, and yield-related parameters under the FL and LM regime conditions (Figure 1, Figure 2 and Figure 3). Additionally, under both irrigation regimes, the treatments consisting of the combination of foliar application and seed soaking with 0.50 mM (S_1_F_1_, S_1_F_2_, and S_1_F_3_) were more effective than those combinations of foliar application and seed soaking with 1.00 mM (S_2_F_1_, S_2_F_2_, and S_2_F_3_) in enhancing growth-, physiological-, and yield-related parameters (Figure 1, Figure 2 and Figure 3). These results indicate that before treating plants with SA as an alternative strategy to enhance wheat performance under either normal or stress conditions, application methods and concentrations of SA should be taken into account to achieve the maximum results from the exogenous application of SA. Unfortunately, there is still debate in previous studies about the appropriate application methods and concentrations of SA for attaining better results from the exogenous application of this plant hormone. For instance, Farooq et al. [28] reported that foliar application of SA was more efficient than seed soaking and that the moderate concentration (100 mg L^−1^) was also more efficient than the low (50 mg L^−1^) and high (150 mg L^−1^) concentrations for enhancing the performance of rice under normal and water-deficit stress conditions. Additionally, Aires et al. [50] confirmed that foliar application of SA is an efficient technique capable of mitigating the negative impacts of water-deficit stress on the production and photosynthetic efficiency of tomato crops. In contrast, the exogenous application of SA was more efficient in improving the growth and production of crops when applied by seed soaking than foliar application at the same concentration [27,51,52]. However, other investigations found that seed soaking and foliar application did not show any significant differences with respect to enhancing plant growth and reducing the negative impacts of water deficit and that low concentrations were more effective than high concentrations [26,53]. The findings of these investigations and our results once again confirm that environmental conditions, plant species, and stress levels might be the main reasons why the application methods and concentrations of SA differ from one study to another. 

Similarly, the PCA and heatmap, which provide an overview of the responses of plant parameters to different SA treatments, showed that the following treatments: F_1_, F_2_, F_3_, S_1_F_1_, S_1_F_2_, and S_1_F_3_ were grouped together and situated along the quarters with the highest PC1 under both irrigation regimes (Figure 4). Additionally, these treatments were clustered into one group under both irrigation regimes and displayed the highest values for all parameters (Figure 5). All of these findings confirm that the foliar application of SA at concentrations of 1–3 mM alone or in combination with seed soaking at a concentration of 0.5 mM seems to be an effective practice for achieving the optimal performance of wheat under arid conditions. These results also confirm that SA application methods are essential factors regarding the effectiveness of SA treatment under control and/or stress conditions. Thus, the foliar application of SA can be suggested as the best method for treating wheat with SA in order to sustain wheat production under similar agro-ecosystem conditions. Similarly, there are some studies that have reported that foliar application of SA is the most effective technique capable of mitigating the negative impacts of environmental stress on the growth and production of field crops [28,50,54]. Since the majority of physiological and biochemical processes and reactions in plants occur in leaves, this may explain why the foliar application of SA seems to be an effective practice for enhancing the growth and production of wheat. Additionally, foliar application of SA may enable adequate accumulations of SA in leaves, helping plants to synthesize/mobilize defense effectors and ensure the quick relief of physiological stress, particularly under stress conditions. The efficiency of the combination of foliar application with seed soaking in this study may indicate that seed soaking, which may induce modification before stress, and foliar application, which may increase the accumulation of SA in leaves during stress, could be required to ensure the growth and production of wheat under stress conditions. Therefore, the application of SA before and during water-deficit stress exposure may also play an important role in mitigating the negative impacts of this kind of stress on the growth and production of wheat. 

Interestingly, the heatmap also revealed that the S_0_ treatment was clustered into one group under the LM regime (Figure 5). This indicates that the exogenous application of SA, regardless of the application method and concentration, plays an efficient role in mitigating the negative impacts of water-deficit stress and improving the growth, production, and IWUE of wheat when grown in regions with low water availability. 

## 4. Materials and Methods

### 4.1. Experimental Site and Conditions 

Two field experiments were performed during the winter growing seasons of 2019/2020 and 2020/2021 to investigate the impact of the application methods and concentrations of SA on physiological, vegetative growth, yield, and water use efficiency attributes of wheat under FL and LM irrigation regimes. The two experiments were implemented at the Research Station of the College of Food and Agriculture Sciences, King Saud University, which is located in the southeastern region of Riyadh City, Saudi Arabia, between latitude 24°24′30″ N and longitude 46°39′30″ E, 400 m above sea level. The experimental area has a typical arid climate, where temperatures range from 10 °C in the winter to 50 °C in the summer, and there is an annual precipitation of approximately 50 mm in the period from the middle of September to the middle of March. Monthly averages of climatic data obtained at the Research Station during wheat growing periods are presented in Figure 6. The soil of the experimental site is sandy loam in texture, with a pH, electrical conductivity, bulk density, organic matter content, water-holding capacity, and permanent wilting point of 7.85, 3.5 dS m^−1^, 1.48 g cm^−3^, 0.46%, 18.89%, and 7.28%, respectively.

### 4.2. Experimental Design and Treatments 

The experiment of this study was laid out in a randomized complete block design with a split-plot arrangement and three replications; irrigation regimes were allocated in the main plots and the twelve treatments of SA were randomly distributed in subplots. Each subplot area was 8.0 m^2^, including 10 rows 4 m in length with 20 cm of spacing between them. The irrigation treatment consisted of FL (100% of the estimated crop evapotranspiration; ETc) and LM (50% ETc) irrigation regimes. The twelve treatments of SA included possible combinations of application methods and concentrations of SA. These treatments included soaking the seeds in purified water (S_0_), 0.5 mM SA (S_1_), and 1.0 mM SA (S_2_); a foliar spray of SA at concentrations of 1.0 mM (F_1_), 2.0 mM (F_2_), and 3.0 mM (F_3_); and combinations of S_1_ and S_2_ with F_1_ (S_1_F_1_ and S_2_F_1_), F_2_ (S_1_F_2_ and S_2_F_2_), and F_3_ (S_1_F_3_ and S_2_F_3_).

The following equation was used to estimate the quantity of irrigation water required for the FL irrigation regime:(1)ETc=ETo×Kc
where ETc, ETo, and Kc are the crop evapotranspiration, reference evapotranspiration, and crop coefficient, respectively. The ETo was estimated using the modified Penman–Monteith equation stated by Allen et al. [55], based on the daily climatic data of the experimental site, such as wind speed, relative humidity, net solar radiation, air temperature, soil heat flux density, saturation and actual vapor pressure, and the slope of the saturation vapor pressure curve. As Kc varies based on local conditions, the Kc values for FAO-56 were adjusted based on the climatic data for wind speed and relative humidity at the study location [55]. Based on the ETc equation, the cumulative of the irrigation amount for the FL regime was approximately 6300 m^3^ ha^−1^. Half of this amount (3150 m^3^ ha^−1^) was applied for the LM regime. 

For the seed soaking treatments, sodium salicylate (2-Hydroxybenzoic acid, HOC6H4COOH) was used to prepare the SA solutions by thoroughly dissolving them in absolute ethanol to make a stock. Thereafter, drops from this stock were added to distilled water to prepare the solutions with 0.5 and 1.0 mM SA. A homogenous lot of wheat seeds was soaked in each concentration of the solution, as well as in distilled water to serve as a control, for 12 h at 25 ± 2 °C at a ratio of 5:1 solution volume to grain weight (v/w). Finally, the soaked seeds were air-dried for 3 h before sowing. 

For the foliar spraying treatments, the SA solutions at concentrations of 1.0, 2.0, and 3.0 mM were prepared as mentioned above with the addition of 0.1% Tween-20 and carefully sprayed on the foliage of wheat plants until the run-off point. To expose all of the plants to the same conditions, the foliage of wheat plants in the control treatment was sprayed with distilled water containing 0.1% Tween-20. The foliar applications were performed twice at tillering and booting growth stages (40 and 60 DAS, respectively) using a backpack Knapsack pressure sprayer (16 L) with a T-jet nozzle that was calibrated to deliver 15 mL s^−1^ at a pressure of 207 kPa. 

### 4.3. Crop Husbandry 

The field used for the experiment was prepared by ploughing of the soil twice before being leveled and divided into subplots (4 m × 2 m each), with a 1 m buffer zone between two adjacent subplots. Then, the treated and untreated seeds of the spring wheat cultivar Summit (*Triticum aestivum* L.) were planted at a seeding rate of 150 kg ha^−1^ on December 5th and 10th in the winter seasons of 2019 and 2020, respectively. Phosphorus fertilizer was applied in one dose at the time of sowing at a rate of 90 kg P_2_O_5_ ha^−1^ in the form of calcium superphosphate (17.0% P_2_O_5_). Potassium fertilizer was applied at two equal doses at the time of sowing and at booting stage at a rate of 60 kg K_2_O ha^−1^ in the form of potassium sulfate (50% K_2_O). Nitrogen fertilizer was applied at three equal doses at seeding, stem elongation, and booting stages at a rate of 180 kg N ha^−1^ in the form of urea (46.5% N). Other agronomic practices, such as removing weeds and protecting plants from diseases, were carried out in a timely manner. 

A low-pressure surface irrigation system was used in this study for the application of irrigation water. This irrigation system consisted of a 76 mm main water plastic pipe, which transferred the water from the main water source to each subplot. This plastic pipe branched off at each subplot into sub-main hoses and was equipped with a manual control valve in order to allow control of the amount of irrigation water delivered to each subplot.

### 4.4. Data Recorded 

#### 4.4.1. Vegetative Growth Parameters

At 80 and 100 DAS, ten representative plants were uprooted from each subplot to measure different traits related to plant growth characteristics, including plant height (PH), tiller number per plant (TN), shoot fresh weight (SFW) and shoot dry weight (SDW) per plant, and green leaf number (GLN) and green leaf area (GLA) per plant. After PH, TN, GLN, and SFW values were recorded, all green leaf blades were separated and their GLAs were measured using a leaf area meter (LI 3100; LI-COR Inc., Lincoln, NE, USA). Subsequently, all parts of the ten plants were dried in a hot-air oven at 75 °C until a constant weight was reached to record SDW. 

Relative leaf water content (RWC) and total chlorophyll content (Chlt) were measured in fully expanded, uppermost leaves of five plants in each subplot at 100 DAS. An area of 10 cm^2^ from each leaf was excised, and their fresh weights (FWs) were immediately recorded. Subsequently, the leaf samples were floated on distilled water at 25 °C for 10 h under low light conditions to obtain turgid weight values (TWs). Finally, the leaf samples were dried in a hot-air oven at 75 °C until a constant weight was reached and weighed to record their dry weights (DWs). The values of the three parameters were applied in the following equation to calculate the RWC:(2)$$\mathrm{RLWC}\;\left(\%\right)=\frac{\mathrm{FW}-\mathrm{DW}}{\mathrm{TW}-\mathrm{DW}}\times 100$$

Fresh leaf samples (200 mg) from each subplot were soaked individually in 5 mL acetone (80%) and kept in the dark at 25 °C until the leaf tissue was completely colorless. Thereafter, the extracted sap was centrifuged at 3000× *g* for 10 min, and the supernatants were used to read absorbance calorimetrically at A645 and A663 nm wavelengths using a spectrophotometer (UV-2550, Shimadzu, Tokyo, Japan). Finally, the concentrations of Chlt (mg g^−1^ fresh weight) were calculated based on the following equation described by Arnon [56] and Lichtenthaler [57]:(3)Cht=[(20.2×A645)+(8.02×A663)]×V/1000×FW
where A, V, and FW indicate the absorbance at a specific wavelength, the final volume of the extract (ml), and the fresh weight of the tissue extracted (g), respectively.

#### 4.4.2. Yield Parameters

Plants in all treatments were harvested on 24 April in both growing seasons. Fifty spikes were randomly sampled from each subplot to estimate the different yield components, namely, spike length (SL), spikelet number per spike (SpNS), grain number per spike (GNS), grain weight per spike (GWS), and 1000-grain weight (TGW). Thereafter, plants from the central area of each subplot (3.0 m^2^) were manually harvested, tied into bundles, sun-dried for five days, and weighed to determine biological yields (BYs). Subsequently, the plants were threshed, and grains were collected, cleaned, adjusted to 15.5% water content, and weighed to determine grain yields (GYs). After the GY and BY for each plant had been expressed as kg ha^−1^, harvest index (HI) and irrigation water use efficiency (IWUE) were calculated by dividing GY with BY and ETc (mm), respectively. 

### 4.5. Data Analysis

Data were statistically analyzed by subjecting the data to the analysis of variance (ANOVA) ideal for the split plot in a randomized complete block design using SAS 9.4 software (SAS Institute, Cary, NC, USA). Duncan’s test was applied for the separation of treatment means, and significant differences were accepted at the levels of *p* < 0.05, 0.01, and 0.001 [58]. Pearson’s correlation analysis was performed to elucidate the degree of correlation between all parameters under the FL and LM regimes. Principal component analysis (PCA) and heatmapping with clustering were performed to integrate growth and yield parameters with different SA treatments, explain the highest proportion of variation among variables, and reduce the dimensionality and complexity of data. A heatmap was produced using the heatmap packages in R statistical software (RStudio Boston, MA; available at: http://www.rstudio.org/ 31 October 2022 PCA was performed using XLSTAT statistical package software (vers. 2019.1, Excel Add-ins soft SARL, New York, NY, USA). All figures were plotted using SigmaPlot 15 software.

## 5. Conclusions

Under the LM regime, all parameters of vegetative growth at 80 and 100 DAS, physiological parameters at 100 DAS, and yield were significantly decreased, while IWUE was increased compared to the FL regime. However, the results of this study confirm the hypothesis that the exogenous application of SA might play a vital role in modulating biochemical and physiological processes, ultimately mitigating the negative impact on these parameters caused by water deficit under field conditions. The results also indicate that the application methods and concentrations of SA are also an important factor for a positive effect of SA on wheat performance under both normal and stress conditions. Multivariate analysis identified the foliar application method with 1–3 mM SA alone or in combination with seed soaking with 0.5 mM SA as the best treatments for the optimal performance of wheat under both irrigation regimes. Overall, the exogenous application of SA, which is readily available, can serve as a promising and practical strategy to prevent wheat losses in areas where water deficit is a major constraint.

## Figures and Tables

**Figure 1 plants-12-01019-f001:**
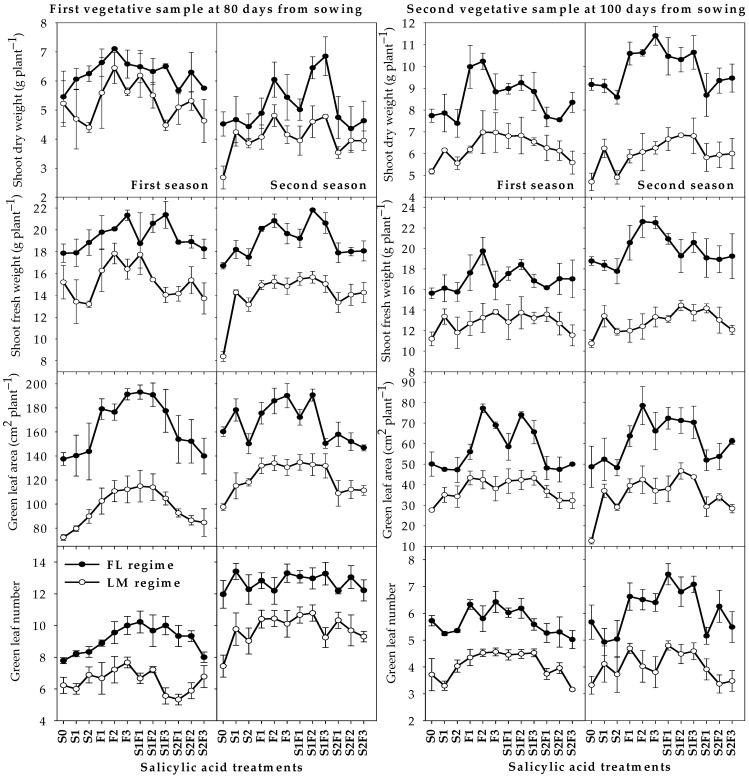
Responses of different vegetative growth parameters of wheat at 80 and 100 days from sowing to the interaction between irrigation regimes and salicylic acid treatments in two growing seasons. Abbreviations in the figure indicate seed soaking in purified water (S_0_), 0.5 mM SA (S_1_), and 1.0 mM SA (S_2_); foliar spray of SA at concentrations of 1.0 mM (F_1_), 2.0 mM (F_2_), and 3.0 mM (F_3_); and combinations of S_1_ and S_2_ with F_1_ (S_1_F_1_ and S_2_F_1_), F_2_ (S_1_F_2_ and S_2_F_2_), and F_3_ (S_1_F_3_ and S_2_F_3_). FL and LM indicate full irrigation regime and limited irrigation regime. The bars are means ± SEs (n = 3).

**Figure 2 plants-12-01019-f002:**
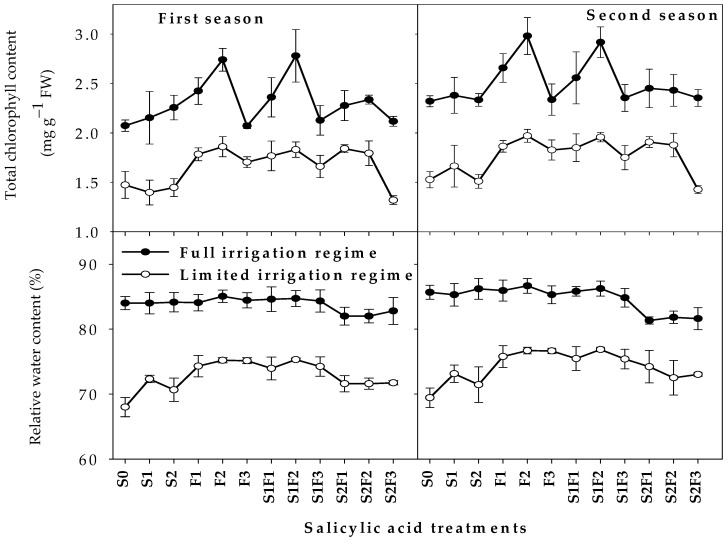
Responses of physiological parameters of wheat at 100 days from sowing to the interaction between irrigation regimes and salicylic acid treatments in two growing seasons. Abbreviations in the figure indicate seed soaking in purified water (S_0_), 0.5 mM SA (S_1_), and 1.0 mM SA (S_2_); foliar spray of SA at concentrations of 1.0 mM (F_1_), 2.0 mM (F_2_), and 3.0 mM (F_3_); and combinations of S_1_ and S_2_ with F_1_ (S_1_F_1_ and S_2_F_1_), F_2_ (S_1_F_2_ and S_2_F_2_), and F_3_ (S_1_F_3_ and S_2_F_3_). The bars are means ± SEs (n = 3).

**Figure 3 plants-12-01019-f003:**
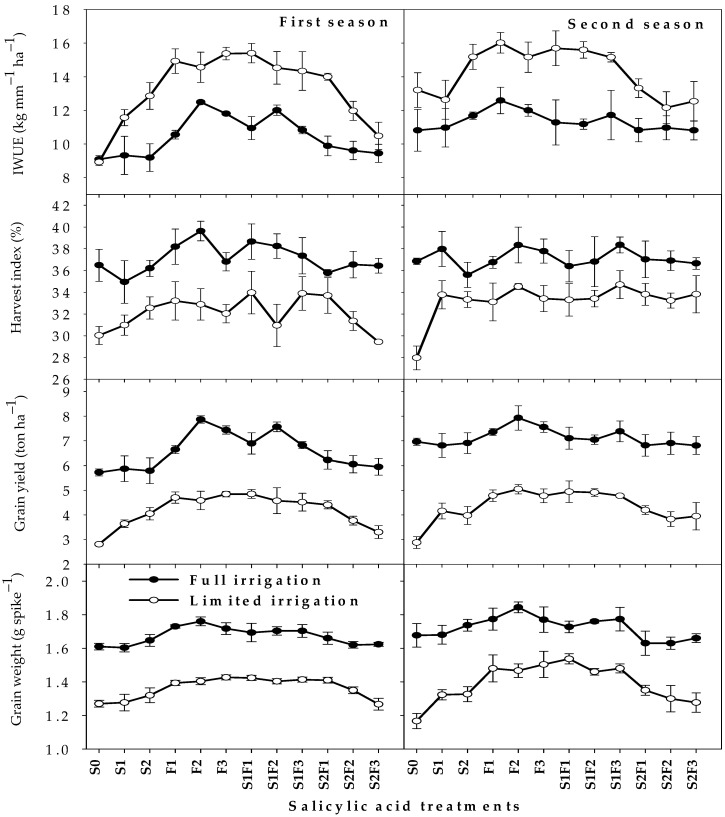
Responses of yield parameters and irrigation water use efficiency (IWUE) of wheat to the interaction between irrigation regimes and salicylic acid treatments in two growing seasons. Abbreviations in the figure indicate seed soaking in purified water (S_0_), 0.5 mM SA (S_1_), and 1.0 mM SA (S_2_); foliar spray of SA at concentrations of 1.0 mM (F_1_), 2.0 mM (F_2_), and 3.0 mM (F_3_); and combinations of S_1_ and S_2_ with F_1_ (S_1_F_1_ and S_2_F_1_), F_2_ (S_1_F_2_ and S_2_F_2_), and F_3_ (S_1_F_3_ and S_2_F_3_). The bars are means ± SEs (n = 3).

**Figure 4 plants-12-01019-f004:**
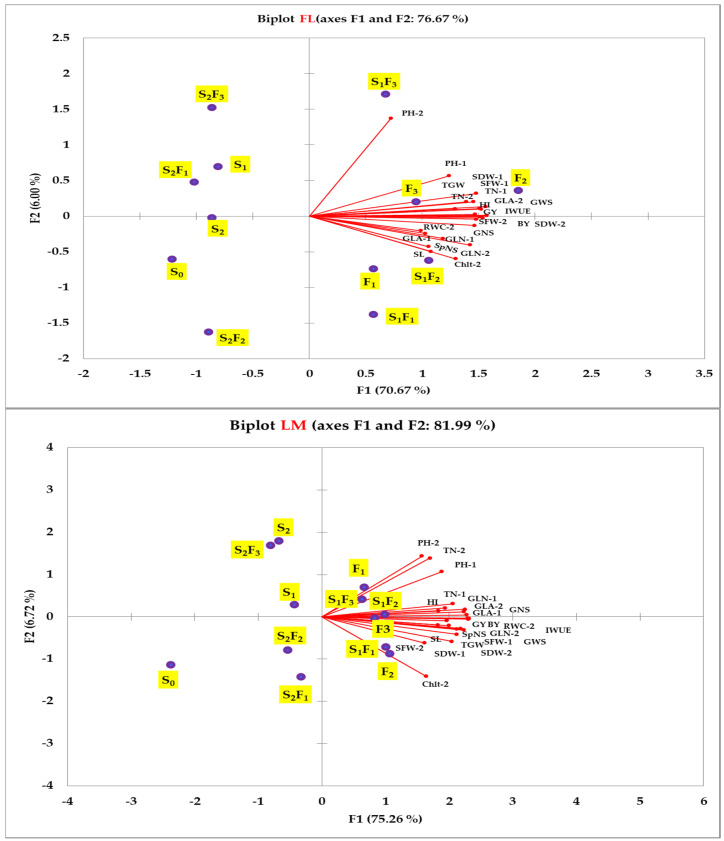
Biplot of principal component analysis for the first two principal components of different parameters of wheat and different salicylic acid treatments under full (FL) and limited (LM) irrigation regimes. Abbreviations in the figure indicate seed soaking in purified water (S_0_), 0.5 mM SA (S_1_), and 1.0 mM SA (S_2_); foliar spray of SA at concentrations of 1.0 mM (F_1_), 2.0 mM (F_2_), and 3.0 mM (F_3_); and combinations of S_1_ and S_2_ with F_1_ (S_1_F_1_ and S_2_F_1_), F_2_ (S_1_F_2_ and S_2_F_2_), and F_3_ (S_1_F_3_ and S_2_F_3_). PH, plant height (cm plant^−1^); TN, tiller number per plant; GLN, green leaf number per plant; GLA, green leaf area (cm^2^ plant^−1^); SFW, shoot fresh weight (g plant^−1^); SDW, shoot dry weight (g plant^−1^); RWC, relative water content (%); Chlt, total chlorophyll content (mg g^−1^ fresh weight), SL, spike length (cm); SpNS, spikelet number per spike; GNS, grain number per spike; GWS, grain weight per spike (g); TGW, thousand-grain weight (g); GY, grain yield (ton ha^−1^); BY, biological yield (ton ha^−1^); HI, harvest index (%); IWUE, irrigation water use efficiency (kg mm^−1^ ha^−1^). The numbers 1 and 2 indicate traits measured at 80 and 100 days after sowing, respectively.

**Figure 5 plants-12-01019-f005:**
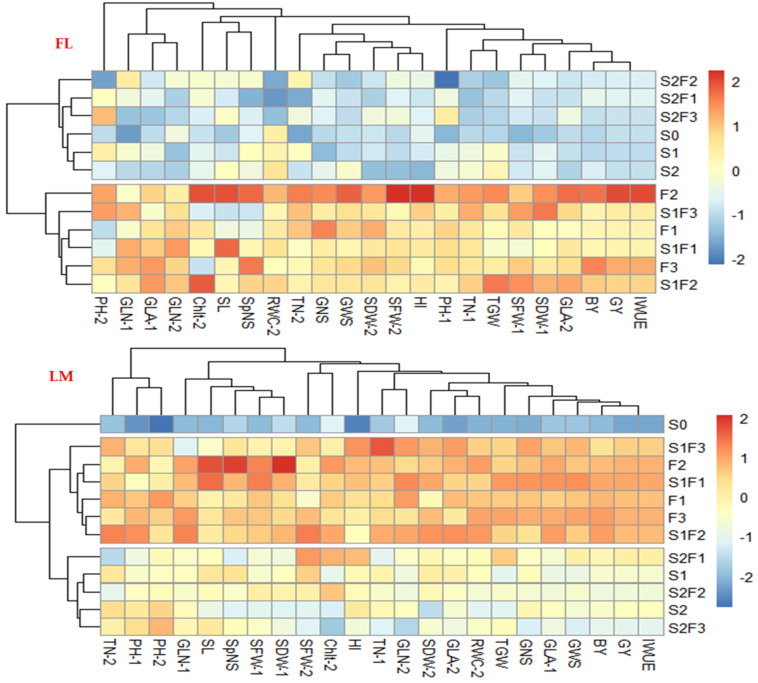
Heatmap of different parameters of wheat and different salicylic acid treatments under full (FL) and limited (LM) irrigation regimes. Abbreviations in the figure indicate seed soaking in purified water (S_0_), 0.5 mM SA (S_1_), and 1.0 mM SA (S_2_); foliar spray of SA at concentrations of 1.0 mM (F_1_), 2.0 mM (F_2_), and 3.0 mM (F_3_); and combinations of S_1_ and S_2_ with F_1_ (S_1_F_1_ and S_2_F_1_), F_2_ (S_1_F_2_ and S_2_F_2_), and F_3_ (S_1_F_3_ and S_2_F_3_). PH, plant height (cm plant^−1^); TN, tiller number per plant; GLN, green leaf number per plant; GLA, green leaf area (cm^2^ plant^−1^); SFW, shoot fresh weight (g plant^−1^); SDW, shoot dry weight (g plant^−1^); RWC, relative water content (%); Chlt, total chlorophyll content (mg g^−1^ fresh weight), SL, spike length (cm); SpNS, spikelet number per spike; GNS, grain number per spike; GWS, grain weight per spike (g); TGW, thousand-grain weight (g); GY, grain yield (ton ha^−1^); BY, biological yield (ton ha^−1^); HI, harvest index (%); IWUE, irrigation water use efficiency (kg mm^−1^ ha^−1^). The numbers 1 and 2 indicate traits measured at 80 and 100 days after sowing, respectively.

**Figure 6 plants-12-01019-f006:**
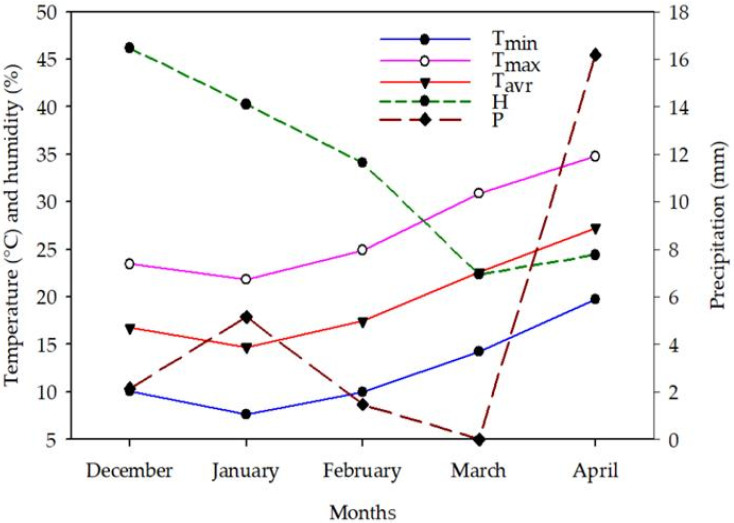
Climatic data for the growing seasons of spring wheat recorded at the Research Station. Al the values for maximum (T_max_), minimum (T_min_), and average (T_avr_) temperatures, humidity (H), and precipitation (P) are the monthly averages for two growing seasons.

**Table 1 plants-12-01019-t001:** Analysis of variances (mean squares) for different vegetative growth parameters (Par.) of wheat at 80 and 100 days from sowing in two growing seasons (n = 3).

Par.	First Season (2019/2020)				Second Season (2020/2021)			
	IR	SA	IR × SA	Error	IR	SA	IR × SA	Error
DF	1	11	11	44	1	11	11	44
**First vegetative sample at 80 days from sowing**								
PH	3007.78 ***	4.31 ^ns^	0.490 ^ns^	2.40	3108.92 *	16.74 *	8.76 ^ns^	8.25
TN	1.78 *	0.406 ^ns^	0.097 ^ns^	0.225	14.81 **	1.74 ***	0.030 ^ns^	0.168
GLN	121.91 **	2.08 ***	1.60 ***	0.314	157.95 **	2.32 ***	1.12 *	0.437
GLA	82,246.9 ***	1988.26 ***	119.48 *	129.55	37,967.1 **	999.54 ***	288.35 ***	66.23
SFW	310.13 **	7.99 ***	4.24 *	1.61	449.80 ***	16.17 ***	2.60 ***	0.665
SDW	17.82 *	1.47 ***	0.458 *	0.240	22.75 *	2.62 ***	0.489 *	0.285
**Second vegetative sample at 100 days from sowing**								
PH	3631.22 ***	4.78 ^ns^	0.571 ^ns^	2.44	3430.13 *	23.19 ^ns^	23.55 ^ns^	12.28
TN	3.43 *	0.189 ^ns^	0.112 ^ns^	0.114	29.79 *	0.585 ^ns^	0.285 ^ns^	0.299
GLN	47.27 ***	1.25 ***	0.153 *	0.070	78.83 **	2.30 ***	0.629 *	0.254
GLA	7277.00 **	351.56 ***	103.48 ***	25.75	12,876.92 **	499.69 ***	72.97 *	33.22
SFW	320.93 *	4.38 ***	2.15 *	1.00	887.54 **	5.95 **	4.78 **	1.74
SDW	94.65 *	2.99 ***	0.784 *	0.319	267.54 ***	3.04 ***	0.787 *	0.369
RWC	2181.19 ***	12.45 ***	5.67 **	1.80	1987.97 **	17.52 ***	10.44 ***	2.12
Chlt	7.68 *	0.234 ***	0.052 **	0.015	9.99 *	0.219 ***	0.043 **	0.014

IR and SA indicate irrigation regimes and salicylic acid treatments, respectively. PH, plant height (cm plant^−1^); TN, tiller number per plant; GLN, green leaf number per plant; GLA, green leaf area (cm^2^ plant^−1^); SFW, shoot fresh weight (g plant^−1^); SDW, shoot dry weight (g plant^−1^); RLWC, relative water content (%); Chlt, total chlorophyll content (mg g^−1^ fresh weight). ns, denotes non-significance, and *, **, and *** indicate significance at *p* ≤ 0.05, 0.01, 0.001 in F-tests, respectively.

**Table 2 plants-12-01019-t002:** Effects of irrigation regime treatments on different growth parameters (Par.) of wheat at 80 and 100 days from sowing in two growing seasons (n = 3).

Par.	First Season (2019/2020)			Second Season (2020/2021)		
	FL	LM	Change (%)	FL	LM	Change (%)
	First Vegetative Sample at 80 Days from Sowing					
PH	84.66 a	71.74 b	15.3	85.79 a	72.65 b	15.3
TN	5.23 a	4.92 b	5.9	4.69 a	3.79 b	19.2
GLN	9.11 a	6.51 b	28.6	12.73 a	9.77 b	23.3
GLA	164.67 a	97.08 b	41.0	167.55 a	121.62 b	27.4
SFW	19.38 a	15.23 b	21.4	19.05 a	14.06 b	26.2
SDW	6.26 a	5.26 b	16.0	5.17 a	4.05 b	21.7
	**Second vegetative sample at 100 days from sowing**					
PH	88.51 a	74.31 b	16.0	88.90 a	75.09 b	15.5
TN	4.88 a	4.44 b	9.0	5.18 a	3.89 b	24.9
GLN	5.69 a	4.07 b	28.5	6.12 a	4.02 b	34.3
GLA	57.65 a	37.43 b	35.1	61.54 a	34.80 b	43.5
SFW	17.03 a	12.81 b	24.8	19.89 a	12.87 b	35.3
SDW	8.56 a	6.26 b	26.9	9.87 a	6.01 b	39.1
RWC	83.85 a	72.84 b	13.1	84.73 a	74.22 b	12.4
Chlt	2.31 a	1.66 b	28.3	2.51 a	1.76 b	29.7

FL, full irrigation regime; LM, limited irrigation regime; PH, plant height (cm plant^−1^); TN, tiller number per plant; GLN, green leaf number per plant; GLA, green leaf area (cm^2^ plant^−1^); SFW, shoot fresh weight (g plant^−1^); SDW, shoot dry weight (g plant^−1^); RLWC, relative water content (%); Chlt, total chlorophyll content (mg g^−1^ fresh weight). Means in the same row designated with different letters indicate significant differences among irrigation treatments at *p* ≤ 0.05 according to Duncan’s test.

**Table 3 plants-12-01019-t003:** Effects of different salicylic acid (SA) treatments on different growth parameters (Par.) of wheat at 80 and 100 days from sowing in two growing seasons (n = 3).

Par.	S_0_	S_1_	S_2_	F_1_	F_2_	F_3_	S_1_F_1_	S_1_F_2_	S_1_F_3_	S_2_F_1_	S_2_F_2_	S_2_F_3_
	First Vegetative Sample at 80 Days from Sowing											
	First Season (2019/2020)											
PH	76.67 ^ns^	77.49 ^ns^	78.43 ^ns^	78.25 ^ns^	79.04 ^ns^	79.19 ^ns^	78.95 ^ns^	78.74 ^ns^	78.18 ^ns^	77.92 ^ns^	76.77 ^ns^	78.77 ^ns^
TN	5.22 ^ns^	5.05 ^ns^	5.17 ^ns^	5.00 ^ns^	5.12 ^ns^	5.17 ^ns^	5.22 ^ns^	5.17 ^ns^	5.61 ^ns^	4.72 ^ns^	4.67 ^ns^	4.78 ^ns^
GLN	7.00 ^d^	7.11 ^d^	7.61 ^cd^	7.78 ^bc^	8.39 ^ab^	8.84 ^a^	8.45 ^a^	8.45 ^a^	7.78 ^bc^	7.33 ^cd^	7.61 ^cd^	7.39 ^cd^
GLA	104.9 ^d^	110.0 ^cd^	116.8 ^bcd^	140.8 ^a^	143.7 ^a^	151.8 ^a^	154.0 ^a^	152.4 ^a^	141.1 ^a^	123.3 ^b^	119.4 ^bc^	112.3 ^bcd^
SFW	16.53 ^cde^	15.66 ^e^	16.02 ^de^	18.03 ^ab^	18.96 ^a^	18.87 ^a^	18.24 ^ab^	18.02 ^ab^	17.72 ^abc^	16.51 ^cde^	17.15 ^bcd^	15.99 ^de^
SDW	5.34 ^de^	5.37 ^cde^	5.33 ^de^	6.11 ^b^	6.78 ^a^	6.10 ^b^	6.34 ^ab^	5.91 ^bc^	5.49 ^cde^	5.38 ^cde^	5.80 ^bcd^	5.19 ^e^
	**Second season (2020/2021)**											
PH	75.37 ^b^	78.61 ^ab^	78.95 ^a^	81.06 ^a^	81.00 ^a^	80.33 ^a^	78.61 ^ab^	80.95 ^a^	80.00 ^a^	77.84 ^ab^	78.06 ^ab^	79.83 ^a^
TN	3.11 ^d^	4.11 ^bc^	4.00 ^c^	4.66 ^a^	4.94 ^a^	4.50 ^ab^	4.50 ^ab^	4.78 ^a^	4.72 ^a^	3.78 ^c^	4.06 ^bc^	3.72 ^c^
GLN	9.70 ^c^	11.59 ^a^	10.65 ^b^	11.62 ^a^	11.31 ^ab^	11.70 ^a^	11.87 ^a^	11.88 ^a^	11.26 ^ab^	11.26 ^ab^	11.37 ^ab^	10.76 ^b^
GLA	128.9 ^e^	146.8 ^bc^	134.2 ^de^	153.8 ^ab^	159.9 ^a^	160.4 ^a^	153.5 ^ab^	161.8 ^a^	141.2 ^cd^	133.5 ^de^	132.0 ^de^	129.1 ^e^
SFW	12.55 ^d^	16.24 ^c^	15.32 ^c^	17.52 ^b^	18.05 ^ab^	17.24 ^b^	17.36 ^b^	18.74 ^a^	17.83 ^ab^	15.62 ^c^	16.03 ^c^	16.16 ^c^
SDW	3.60 ^d^	4.45 ^bc^	4.15 ^cd^	4.48 ^bc^	5.43 ^a^	4.79 ^b^	4.48 ^bc^	5.52 ^a^	5.81 ^a^	4.14 ^cd^	4.16 ^cd^	4.29 ^bc^
	**Second vegetative sample at 100 days from sowing**											
	**First season (2019/2020)**											
PH	79.79 ^ns^	80.72 ^ns^	81.80 ^ns^	81.19 ^ns^	82.48 ^ns^	82.41 ^ns^	81.98 ^ns^	82.04 ^ns^	81.38 ^ns^	81.14 ^ns^	79.94 ^ns^	82.02 ^ns^
TN	4.57 ^ns^	4.65 ^ns^	4.61 ^ns^	4.87 ^ns^	4.81 ^ns^	4.79 ^ns^	4.83 ^ns^	4.77 ^ns^	4.70 ^ns^	4.28 ^ns^	4.40 ^ns^	4.62 ^ns^
GLN	4.72 ^c^	4.27 ^de^	4.69 ^c^	5.34 ^ab^	5.16 ^b^	5.49 ^a^	5.24 ^ab^	5.34 ^ab^	5.06 ^b^	4.50 ^cd^	4.63 ^c^	4.09 ^e^
GLA	38.84 ^d^	41.24 ^d^	40.73 ^d^	49.64 ^c^	59.76 ^a^	53.50 ^bc^	50.17 ^c^	58.12 ^ab^	54.43 ^abc^	42.40 ^d^	39.86 ^d^	41.08 ^d^
SFW	13.42 ^e^	14.75 ^cd^	13.78 ^de^	15.15 ^bc^	16.50 ^a^	15.11 ^bc^	15.19 ^bc^	16.09 ^ab^	15.04 ^bc^	14.87 ^cd^	14.87 ^cd^	14.29 ^cde^
SDW	6.46 ^c^	7.01 ^c^	6.48 ^c^	8.09 ^ab^	8.60 ^a^	7.90 ^b^	7.89 ^b^	8.04 ^ab^	7.70 ^b^	6.97 ^c^	6.84 ^c^	6.97 ^c^
RWC	76.00 ^d^	78.16 ^bc^	77.40 ^cd^	79.18 ^ab^	80.12 ^a^	79.79 ^a^	79.28 ^ab^	80.01 ^a^	79.29 ^ab^	76.80 ^c d^	76.80 ^cd^	77.27 ^cd^
Chlt	1.77 ^cd^	1.78 ^cd^	1.85 ^cd^	2.10 ^b^	2.30 ^a^	1.89 ^c^	2.06 ^b^	2.31 ^a^	1.89 ^c^	2.06 ^b^	2.07 ^b^	1.72 ^d^
	**Second season (2020/2021)**											
PH	77.06 ^ns^	82.44 ^ns^	81.72 ^ns^	82.95 ^ns^	82.39 ^ns^	83.11 ^ns^	80.72 ^ns^	82.00 ^ns^	84.50 ^ns^	81.89 ^ns^	80.72 ^ns^	84.45 ^ns^
TN	3.78 ^ns^	4.61 ^ns^	4.50 ^ns^	4.77 ^ns^	4.72 ^ns^	4.50 ^ns^	4.50 ^ns^	4.83 ^ns^	4.94 ^ns^	4.16 ^ns^	4.61 ^ns^	4.50 ^ns^
GLN	4.50 ^f^	4.52 ^ef^	4.38 ^f^	5.66 ^abc^	5.27 ^bcd^	5.10 ^cde^	6.13 ^a^	5.64 ^abc^	5.83 ^ab^	4.54 ^ef^	4.81 ^def^	4.48 ^f^
GLA	30.67 ^d^	44.66 ^c^	38.66 ^c^	51.63 ^b^	60.45 ^a^	51.59 ^b^	55.16 ^ab^	58.98 ^a^	57.05 ^ab^	40.60 ^c^	43.80 ^c^	44.77 ^c^
SFW	14.76 ^d^	15.89^cd^	14.84 ^d^	16.28 ^bcd^	17.51 ^ab^	17.92 ^a^	17.03 ^abc^	16.87 ^ab^	17.16 ^ab^	16.62 ^abc^	15.98 ^bcd^	15.66 ^cd^
SDW	6.94 ^e^	7.67 ^bcd^	6.75 ^e^	8.23 ^abc^	8.35 ^ab^	8.83 ^a^	8.56 ^a^	8.58 ^a^	8.73 ^a^	7.25 ^de^	7.64 ^cd^	7.73 ^bcd^
RWC	77.56 ^de^	79.21 ^bcd^	78.83 ^cde^	80.86 ^a^	81.68 ^a^	80.97 ^a^	80.64 ^ab^	81.55 ^a^	80.11 ^bc^	77.77 ^de^	77.17 ^e^	77.32 ^e^
Chlt	1.92 ^fg^	2.02 ^efg^	1.92 ^fg^	2.26 ^b^	2.48 ^a^	2.08 ^cde^	2.20 ^bc^	2.44 ^a^	2.05 ^def^	2.18 ^bcd^	2.15 ^bcde^	1.89 ^g^

The treatments included seed soaking in purified water (S_0_), 0.5 mM SA (S_1_), and 1.0 mM SA (S_2_); foliar spray of SA at concentrations of 1.0 mM (F_1_), 2.0 mM (F_2_), and 3.0 mM (F_3_); and combinations of S_1_ and S_2_ with F_1_ (S_1_F_1_ and S_2_F_1_), F_2_ (S_1_F_2_ and S_2_F_2_), and F_3_ (S_1_F_3_ and S_2_F_3_). PH, plant height (cm plant^−1^); TN, tiller number per plant; GLN, green leaf number per plant; GLA, green leaf area (cm2 plant^−1^); SFW, shoot fresh weight (g plant^−1^); SDW, shoot dry weight (g plant^−1^); RLWC, relative water content (%); Chlt, total chlorophyll content (mg g^−1^ fresh weight). Means in the same row designated with different letters indicate significant differences among SA treatments at *p* ≤ 0.05 according to Duncan’s test. ns, denotes non-significance

**Table 4 plants-12-01019-t004:** Analysis of variances (mean squares) for different yield parameters (Par.) and irrigation water use efficiency of wheat in two growing seasons.

Par.	First Season (2019/2020)				Second Season (2020/2021)			
	IR	SA	IR × SA	Error	IR	SA	IR × SA	Error
DF	1	11	11	44	1	11	11	44
SL	13.09 **	0.472 **	0.052 ^ns^	0.155	2.27 **	0.100 *	0.039 ^ns^	0.048
SpNS	8.27 *	0.148 ^ns^	0.111 ^ns^	0.170	7.61 **	0.379 **	0.162 ^ns^	0.122
GNS	312.38 **	4.30 *	0.577 ^ns^	2.07	449.60 **	11.54 ***	4.63 ^ns^	1.29
GWS	1.72 **	0.018 ***	0.002 *	7.82	1.99 ***	0.042 ***	0.010 **	0.003
TGW	262.63 **	3.07 **	1.66 ^ns^	0.925	230.73 **	4.74 ***	1.87 ^ns^	1.26
GY	103.58 ***	2.63 ***	0.361 **	0.099	139.11 **	1.23 ***	0.375 *	0.169
BY	400.59 ***	14.34 ***	1.35 ^ns^	0.700	680.56 ***	6.91 ***	2.10 ^ns^	1.09
HI	455.42 **	8.59 ***	3.86 *	1.81	276.75 *	6.81 **	4.16 *	2.40
IWUE	142.58 **	14.40 ***	2.98 ***	0.530	112.15 *	8.81 ***	4.51 ***	0.871

FL, full irrigation regime; LM, limited irrigation regime; SL, spike length (cm); SpNS, spikelet number per spike; GNS, grain number per spike; GWS, grain weight per spike (g); TGW, thousand-grain weight (g); GY, grain yield (ton ha^−1^); BY, biological yield (ton ha^−1^); HI, harvest index (%); IWUE, irrigation water use efficiency (kg mm^−1^ ha^−1^). ns, denotes non-significance, and *, **, and *** indicate significance at *p* ≤ 0.05, 0.01, and 0.001 in the F-test, respectively.

**Table 5 plants-12-01019-t005:** Effects of irrigation regimes and different salicylic acid (SA) treatments on different yield parameters and irrigation water use efficiency of wheat in two growing seasons (n = 3).

Treatments	SL	SpNS	GNS	GWS	TGW	GY	BY	HI	IWUE
	First season (2019/2020)								
S_0_	8.17 ^c^	16.28 ^ns^	42.83 ^c^	1.44 ^d^	32.67 ^d^	4.27 ^e^	12.55 ^f^	33.26 ^de^	9.00 ^g^
S_1_	8.52 ^bc^	16.40 ^ns^	43.66 ^bc^	1.44 ^d^	33.56 ^cd^	4.76 ^d^	14.28 ^e^	32.96 ^e^	10.44 ^ef^
S_2_	8.48 ^bc^	16.12 ^ns^	44.29 ^abc^	1.48 ^c^	33.41 ^cd^	4.92 ^d^	14.22 ^e^	34.38 ^cde^	11.02 ^e^
F_1_	8.68 ^b^	16.58 ^ns^	45.58 ^a^	1.56 ^ab^	34.17 ^abc^	5.68 ^bc^	15.81 ^c^	35.70 ^abc^	12.74 ^bcd^
F_2_	9.20 ^a^	16.68 ^ns^	45.20 ^ab^	1.58 ^a^	34.91 ^ab^	6.23 ^a^	16.91 ^ab^	36.26 ^ab^	13.53 ^ab^
F_3_	8.67 ^b^	16.42 ^ns^	44.94 ^ab^	1.57 ^a^	34.73 ^ab^	6.14 ^a^	17.67 ^a^	34.42 ^cde^	13.59 ^a^
S_1_F_1_	9.15 ^a^	16.52 ^ns^	45.56 ^a^	1.56 ^ab^	34.18 ^abc^	5.87 ^ab^	16.06 ^bc^	36.31 ^a^	13.17 ^abc^
S_1_F_2_	8.75 ^ab^	16.43 ^ns^	44.85 ^ab^	1.55 ^ab^	35.16 ^a^	6.07 ^a^	17.29 ^a^	34.60 ^cd^	13.27 ^abc^
S_1_F_3_	8.53 ^bc^	16.43 ^ns^	45.52 ^a^	1.56 ^ab^	34.17 ^abc^	5.67 ^bc^	15.80 ^c^	35.62 ^abc^	12.59 ^cd^
S_2_F_1_	8.63 ^b^	16.22 ^ns^	44.76 ^ab^	1.54 ^b^	34.30 ^abc^	5.32 ^c^	15.25 ^cd^	34.74 ^bcd^	11.94 ^d^
S_2_F_2_	8.55 ^bc^	16.37 ^ns^	43.94 ^abc^	1.49 ^c^	33.79 ^bc^	4.91 ^d^	14.30 ^de^	33.95 ^de^	10.79 ^ef^
S_2_F_3_	8.62 ^bc^	16.28 ^ns^	44.39 ^abc^	1.45 ^d^	33.35 ^cd^	4.63 ^de^	13.78 ^e^	32.93 ^e^	9.96 ^f^
FL	9.09 a	16.73 a	46.71 a	1.67 a	35.94 a	6.57 a	17.68 a	37.11 a	10.43 b
LM	8.24 b	16.06 b	42.54 b	1.36 b	32.12 b	4.17 b	12.97 b	32.08 b	13.24 a
**Change (%)**	**9.4**	**4.0**	**8.9**	**18.6**	**10.6**	**36.5**	**26.7**	**13.6**	**−27.0**
	**Second season (2020/2021)**								
S_0_	8.55 ^c^	16.65 ^d^	42.66 ^d^	1.42 ^d^	33.07 ^d^	4.92 ^e^	14.59 ^c^	32.41 ^c^	10.09 ^c^
S_1_	8.77 ^abc^	17.07 ^abc^	44.06 ^c^	1.50 ^bc^	34.01 ^cd^	5.49 ^d^	15.14 ^c^	35.87 ^ab^	12.01 ^b^
S_2_	8.72 ^abc^	16.97 ^bcd^	44.41 ^bc^	1.53 ^b^	34.36 ^abc^	5.44 ^d^	15.65 ^bc^	34.46 ^b^	11.80 ^b^
F_1_	8.63 ^bc^	16.77 ^cd^	46.04 ^a^	1.63 ^a^	35.26 ^abc^	6.07 ^ab^	17.24 ^a^	34.93 ^ab^	13.43 ^a^
F_2_	8.92 ^a^	17.40 ^a^	46.48 ^a^	1.66 ^a^	35.49 ^ab^	6.49 ^a^	17.65 ^a^	36.43 ^a^	14.31 ^a^
F_3_	8.83 ^ab^	17.38 ^a^	46.25 ^a^	1.64 ^a^	35.34 ^ab^	6.17 ^ab^	17.16 ^a^	35.60 ^ab^	13.58 ^a^
S_1_F_1_	8.90 ^a^	17.08 ^abc^	45.87 ^a^	1.63 ^a^	35.55 ^a^	6.03 ^ab^	17.24 ^a^	34.85 ^ab^	13.49 ^a^
S_1_F_2_	8.72 ^abc^	17.20 ^ab^	45.48 ^ab^	1.61 ^a^	35.34 ^ab^	5.98 ^bc^	16.94 ^a^	35.11 ^ab^	13.39 ^a^
S_1_F_3_	8.59 ^bc^	16.87 ^bcd^	45.64 ^ab^	1.63 ^a^	35.58 ^a^	6.08 ^ab^	16.50 ^ab^	36.52 ^a^	13.44 ^a^
S_2_F_1_	8.54 ^c^	16.65 ^d^	43.46 ^cd^	1.49 ^bc^	34.23 ^bcd^	5.51 ^cd^	15.45 ^bc^	35.42 ^ab^	12.07 ^b^
S_2_F_2_	8.74 ^abc^	16.88 ^bcd^	43.76 ^cd^	1.47 ^cd^	33.37 ^d^	5.37 ^de^	15.12 ^c^	35.07 ^ab^	11.56 ^b^
S_2_F_3_	8.82 ^ab^	16.92 ^bcd^	42.72 ^d^	1.47 ^cd^	34.24 ^bcd^	5.38 ^de^	15.14 ^c^	35.23 ^ab^	11.68 ^b^
FL	8.90 a	17.31 a	47.23 a	1.72 a	36.44 a	7.13 a	19.23 a	37.12 a	11.32 b
LM	8.55 b	16.66 b	42.24 b	1.39 b	32.86 b	4.35 b	13.08 b	33.20 b	13.82 a
**Change (%)**	**3.9**	**3.8**	**10.6**	**19.3**	**9.8**	**39.0**	**32.0**	**10.6**	**−22.0**

The treatments included seed soaking in purified water (S_0_), 0.5 mM SA (S_1_), and 1.0 mM SA (S_2_); foliar spray of SA at concentrations of 1.0 mM (F_1_), 2.0 mM (F_2_), and 3.0 mM (F_3_); and combinations of S_1_ and S_2_ with F_1_ (S_1_F_1_ and S_2_F_1_), F_2_ (S_1_F_2_ and S_2_F_2_), and F_3_ (S_1_F_3_ and S_2_F_3_). SL, spike length (cm); SpNS, spikelet number per spike; GNS, grain number per spike; GWS, grain weight per spike (g); TGW, thousand-grain weight (g); GY, grain yield (ton ha^−1^); BY, biological yield (ton ha^−1^); HI, harvest index (%); IWUE, irrigation water use efficiency (kg mm^−1^ ha^−1^); FL, full irrigation regime; LM, limited irrigation regime. Means in the same column designated with different letters indicate significant differences among treatments at *p* ≤ 0.05 according to Duncan’s test.

## Data Availability

All data are presented within the article.

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
