# Peer review of "Integrating Application Methods and Concentrations of Salicylic Acid as an Avenue to Enhance Growth, Production, and Water Use Efficiency of Wheat under Full and Deficit Irrigation in Arid Countries"

_plants, 2023, doi:10.3390/plants12051019_

Round 1

Reviewer 1 Report

The manuscript aims to evaluate the effects of SA application on wheat growth, yield, and WUE under full and deficit irrigation. The optimal application of SA was developed with the 2-year experiment. The results were very specific to the study region. It is well structured, and the objectives are clear.

Minor points:

1. Hypothesis should be stated clearly in the section of Introduction.

2. The resolution of figures are too low, high resolution figures are required.

3. Discussion should be improved. Do not repeat results in the Discussion section.

3. Weakness of this study should be pointed out. 

Author Response

Reviewer   #1

The manuscript aims to evaluate the effects of SA application on wheat growth, yield, and WUE under full and deficit irrigation. The optimal application of SA was developed with the 2-year experiment. The results were very specific to the study region. It is well structured, and the objectives are clear.

Response: We greatly appreciate your critical observations as well as your constructive and helpful comments. We hope that we could address your questions/comments by the explanations and revisions made in the manuscript. We believe that the manuscript is substantially improved after making the suggested revisions.

Minor points:

  1. Hypothesis should be stated clearly in the section of Introduction.

Response: Thank you very much for your comment. The main hypothesis of the study has been stated clearly in the Introduction section.

  1. The resolution of the figures are too low, high resolution figures are required.

Response: Thank you very much for your comment. High-resolution Figures have been provided.

  1. Discussion should be improved. Do not repeat results in the Discussion section.

Response: Thank you very much for your comment. The Discussion section has been improved and the repeated results in this section have been deleted.

  1. Weakness of this study should be pointed out. 

Response: Thank you very much for your comment. Although there are many studies have been carried out to investigate the effectiveness and economics of salicylic acid for mitigating water deficit in plants, most of these studies were carried out in pot experiments in greenhouse conditions. Most importantly, there is still debate in previous studies about the appropriate application methods and concentrations of SA for getting better results from the exogenous application of this plant hormone.

Reviewer 2 Report

The authors aimed to identify the proper application methods of salicylic acid to investigate its effect on the growth and yield characteristics of wheat under full irrigation or limited irrigation regime in an arid agro-ecosystem. In my opinion, this manuscript is well written, although the Introduction and the References are not formatted according to the journal’s criteria.

·       According to the journal’s criteria, the abstract should be a total of about 200 words maximum. Please shorten it.

·       Please mention the number of the replication by the tables or figures.

·       Please mention the name of the investigated wheat cultivar.

·       In scientific papers, it is customary to give the scientific name of the examined species.

·       Please format the References chapter according to the journal’s criteria. 

Author Response

Reviewer   #2

The authors aimed to identify the proper application methods of salicylic acid to investigate its effect on the growth and yield characteristics of wheat under full irrigation or limited irrigation regime in an arid agro-ecosystem. In my opinion, this manuscript is well written, although the Introduction and the References are not formatted according to the journal’s criteria.

Response: We greatly appreciate your critical observations as well as your constructive and helpful comments. We hope that we could address your questions/comments through the explanations and revisions made in the manuscript. We believe that the manuscript is substantially improved after making the suggested revisions.

  • According to the journal’s criteria, the abstract should be a total of about 200 words maximum. Please shorten it.

Response: Thank you very much for your comment. The abstract section has been shortened.

  • Please mention the number of the replication by the tables or figures. Please mention the name of the investigated wheat cultivar.

Response: Thank you very much for your suggestion. The number of replication has been provided in Tables and Figures. The name of the investigated wheat cultivar has been provided under the subtitle Crop Husbandry. The spring wheat cultivar used in this study is Summit (Triticum aestivum L.).

  • In scientific papers, it is customary to give the scientific name of the examined species.

Response: Thank you very much for your comment. The scientific name has been provided under the subtitle Crop Husbandry. The spring wheat cultivar used in this study is Summit (Triticum aestivum L.).

  • Please format the References chapter according to the journal’s criteria.

Response: Thank you very much for your comment. The References section has been changed according to the journal’s criteria.
